# A deep learning sex-specific body composition ageing biomarker using dual-energy X-ray absorptiometry scan
Jie Lian [1], Pei Cai[1], Fan Huang [1,2], Jianpan Huang[1] & Varut Vardhanabhuti [1,2] ✉

## Abstract

**Background** Chronic diseases are closely linked to alterations in body composition, yet there is a need for reliable biomarkers to assess disease risk and progression. This study aimed to develop and validate a biological age indicator based on body composition derived from dual-energy X-ray absorptiometry (DXA) scans, offering a novel approach to evaluating health status and predicting disease outcomes.

**Methods** A deep learning model was trained on a reference population from the UK Biobank to estimate body composition biological age (BCBA). The model's performance was assessed across various groups, including individuals with typical and atypical body composition, those with pre-existing diseases, and those who developed diseases after DXA imaging. Key metrics such as c-index were employed to examine BCBA's diagnostic and prognostic potential for type 2 diabetes, major adverse cardiovascular events (MACE), atherosclerotic cardiovascular disease (ASCVD), and hypertension.

**Results** Here we show that BCBA strongly correlates with chronic disease diagnoses and risk prediction. BCBA demonstrated significant associations with type 2 diabetes (odds ratio 1.08 for females and 1.04 for males, $p < 0.0005$), MACE (odds ratio 1.10 for females and 1.11 for males, $p < 0.0005$), ASCVD (odds ratio 1.07 for females and 1.10 for males, $p < 0.0005$), and hypertension (odds ratio 1.06 for females and 1.04 for males, $p < 0.0005$). It outperformed standard cardiovascular risk profiles in predicting MACE and ASCVD.

**Conclusions** BCBA is a promising biomarker for assessing chronic disease risk and progression, with potential to improve clinical decision-making. Its integration into routine health assessments could aid early disease detection and personalised interventions.

## Plain Language Summary

Chronic diseases, like diabetes and heart disease, are linked to changes in body composition, such as the amount of fat and muscle in the body. This study aimed to create a new health measure called body composition biological age (BCBA), which uses whole-body x-ray imaging to estimate a person's biological age. A computer model was developed using data from thousands of people in a voluntary database to measure BCBA for different groups, including healthy people and those with or at risk of diseases. They found that BCBA was strongly linked to the likelihood of developing diseases like diabetes and high blood pressure. This study suggests that BCBA could help doctors identify people at higher risk of chronic diseases earlier using this relatively inexpensive tool in a regular health check, allowing for more personalised care and better health outcomes.

The rising worldwide population of aged individuals has led to a large increase in the occurrence of age-related diseases and disorders, including cardiovascular disease, cancer, and dementia[1], posing significant challenges to public health[2,3]. In clinical settings, the risk of an individual developing a disease is commonly evaluated by assessing their chronological age (CA), which represents the duration of time that has passed since their birth[4,5]. However, the process of ageing can vary significantly between individuals[3,6], which can result in the inability of CA to accurately detect age-related diseases. For example, some individuals can preserve their physical and mental health as they get older, while others with accelerated ageing phenotypes may require medical treatment at a younger age[7].

To accurately evaluate the ageing process, it is crucial to establish an accurate biomarker for ageing. To this end, the concept of the biological age (BA) has been proposed to more accurately reflect one's ageing process[8]. BA assesses individuals' biological functioning by comparing it to the expected level for a given chronological age[5,9]. Over the past year, several distinct BA biomarkers have been introduced[10–14]. These biomarkers have the potential to be used as indicators of disease and mortality, providing significant information on the ageing process[15–17]. However, a major barrier to the widespread implementation of these biomarkers is their availability, cost as well as meaningful and actionable insights[18]. Methods such as gene sequencing and Magnetic Resonance Imaging (MRI) scanning, which are essential for many BA examinations, are expensive, making them less

[1]Department of Diagnostic Radiology, Li Ka Shing Faculty of Medicine, The University of Hong Kong, Hong Kong, Hong Kong SAR, China. [2]Snowhill Science Ltd, Units 801-803, Level 8, Core C, Hong Kong, Hong Kong SAR, China. ✉e-mail: varv@hku.hk

accessible to the general public. Whilst several blood-based biomarkers are available[15–17] they often lack explanation with no link to actionable or physiological correlate, making it difficult to design interventions to reduce accelerated ageing, for example[18]. In order to effectively utilise the potential of BA biomarkers in public health for disease risk stratification and prediction, it is crucial to create accurate, cost-effective and physiologically actionable measures that can have physiological and actional insights as alternatives.

Body composition refers to the distribution of adipose tissue and skeletal muscle among individuals. Recent studies have shown that there is a connection between body composition and certain chronic diseases across different cohorts[19–25]. Specifically, these studies highlight the relationship between fat distribution, muscle mass, and the development of chronic conditions such as kidney disease, metabolic disorders, and the developmental origins of non-communicable diseases. For instance, research indicates that the amount of adipose tissue is a reliable indicator of cardiometabolic disorders[21]. Additionally, visceral adipose tissue (VAT) has been shown to be a significant determinant of disease progression in patients with Crohn's disease[22]. Furthermore, individuals who accumulate a higher percentage of fat relative to lean muscle are at an increased risk of developing insulin resistance and metabolic syndrome[21,23].

MRI, dual-energy X-ray absorptiometry (DXA), and bioelectrical impedance analysis are widely used techniques for assessing an individual's body composition. MRI is the most precise option, although it is also the most time-consuming and costly. Although body impedance is readily accessible and affordable, it lacks accuracy at scale[26]. DXA provides accurate results while also being efficient in terms of time and cost. In this regard, body composition biomarkers based on DXA can be readily accessible to the general public. While there have been some studies on body composition using DXA[27], most of them have focused on patients with a single disease for association studies[28] or were conducted with small sample size[29]. As a result, few studies have attempted to routinely employ DXA as a biomarker for ageing and body composition in a large population.

Recently, there has been a significant increase in the use of deep learning (DL) in the field of medical imaging[30,31]. This is due to its excellent ability to identify important latent features behind pixel space. In prior work, the utilisation of DL techniques in medical imaging for building BA models has been shown to be beneficial, particularly with MRI and chest X-ray[7,32]. However, the potential for exploring the relationship between DXA and BA through the analysis of body composition has not previously been investigated.

To address the research gap, we specifically focused on the relationship between body composition and BA, particularly using lean mass and VAT to select eligible candidates. A study involving a large cohort from the UK Biobank found that greater VAT mass was significantly associated with older age[33,34]. Additionally, research examining body composition across various age groups identified breakpoints in the relationships between body composition parameters (including lean mass and fat mass) and age, suggesting that significant associations exist and vary across different age cohorts[34]. Furthermore, a comprehensive review of body composition changes with ageing emphasises that both VAT and lean mass are critical components influencing health risks and functional abilities in older adults[35]. As a result, we believe that lean mass and VAT can be regarded as essential criteria for selecting eligible candidates, allowing us to further explore those candidates' DXA information to build a new BA biomarker.

To achieve this, we proposed a DXA-based body composition biological age (BCBA) by training a deep learning model using a selected population of eligible UK Biobank individuals. We reasoned that the BCBA of the normal reference group would be closer to their CAs, while the BCBA of the disease group would deviate further from their CAs. We applied the established model to patient groups with different chronic diseases to examine the relationship between our proposed BCBA and each condition, evaluating the performance of our BCBA as biomarkers and prognosticators for these diseases.

## Methods

### Study Design

Our analysis focused specifically on Instance 2 of the UK Biobank, which contains data from 3 imaging centres (Cheadle, Newcastle, and Reading). We applied several inclusion criteria. Firstly, individuals are required to have available DXA scan data. Secondly, we restricted our sample to individuals of Caucasian genetic ethnic grouping, aiming to minimise potential confounding variables related to genetic diversity. Thirdly, we ensured that self-reported gender aligned with genetic gender, ensuring consistency and accuracy in gender identification. Finally, we excluded individuals with any self-reported or inpatient records of cancer, as these conditions could significantly impact body composition, potentially confounding our analyses.

Subsequently, from the qualified participants, we selected two disease groups of participants based on their disease status relative to the timing of DXA scans. Specifically, individuals with a documented history of our target diseases prior to the DXA scan were classified into the pre-existing disease group. Conversely, individuals diagnosed with these diseases after the DXA scan were categorised into the post-DXA disease group. After delineating the pre-existing disease and post-DXA disease groups, we further refined our candidates by excluding participants lacking official body composition measurements, specifically VAT and total lean segmentation mass, available in the UK Biobank dataset.

Subsequently, we establish a standard reference subgroup from the remaining individuals by defining them as having VAT and total lean mass one standard deviation from the mean for their respective genders. Participants who met these criteria were categorised as the normal reference group. This group represents individuals in our cohort who had typical body composition profiles. In contrast, those who were outside this range were classified as the non-typical body composition group. Furthermore, the non-typical participants were subsequently categorised into three groups: (1) The Hypernormal group, consisting of individuals with lower visceral adipose tissue (VAT) and higher lean mass percentage (please refer Supplementary Method 1 for detailed percentage calculation) compared to the normal reference group, (2) The Suboptimal group, comprising participants with higher VAT and lower lean mass percentage than the normal reference group, and (3) The Mixed normal group, encompassing the remaining participants. This comprehensive categorisation allows for comparative evaluations of individuals with varying body composition phenotypes. Please refer to Fig. 1 for a comprehensive overview of the study design.

This study encompasses four diseases/outcomes of interest: type 2 diabetes mellitus (T2DM), atherosclerotic cardiovascular disease (ASCVD), hypertension, and major adverse cardiovascular events (MACE). These were selected mainly due to their significant prevalence, large number of positive cases in this dataset, and impact on public health, particularly in relation to body composition and metabolic health.

### Participants

In our study, we utilised data from the UK Biobank, an observational study comprising more than 500,000 individuals across four data collection instances (Application Number 78730). The study protocol was approved by The University of Hong Kong Ethics Review Board (UW-20814) with a waiver of the requirement for informed consent due to its retrospective nature.

### Target Disease

To determine the occurrence of diseases in our group, we employed various sources of information, such as ICD-10 codes, self-reported data collected through verbal interviews conducted by healthcare professionals who received specialized training, records from general practitioners, and national death registries. For detailed information on the UK Biobank data field and ICD-10 Codes, please refer to Supplementary Method 2.

### Body Composition Ageing Model Building

To describe a normal population's body composition, we train a Densenet-121[36] model utilising transfer learning (ImageNet) to calculate an

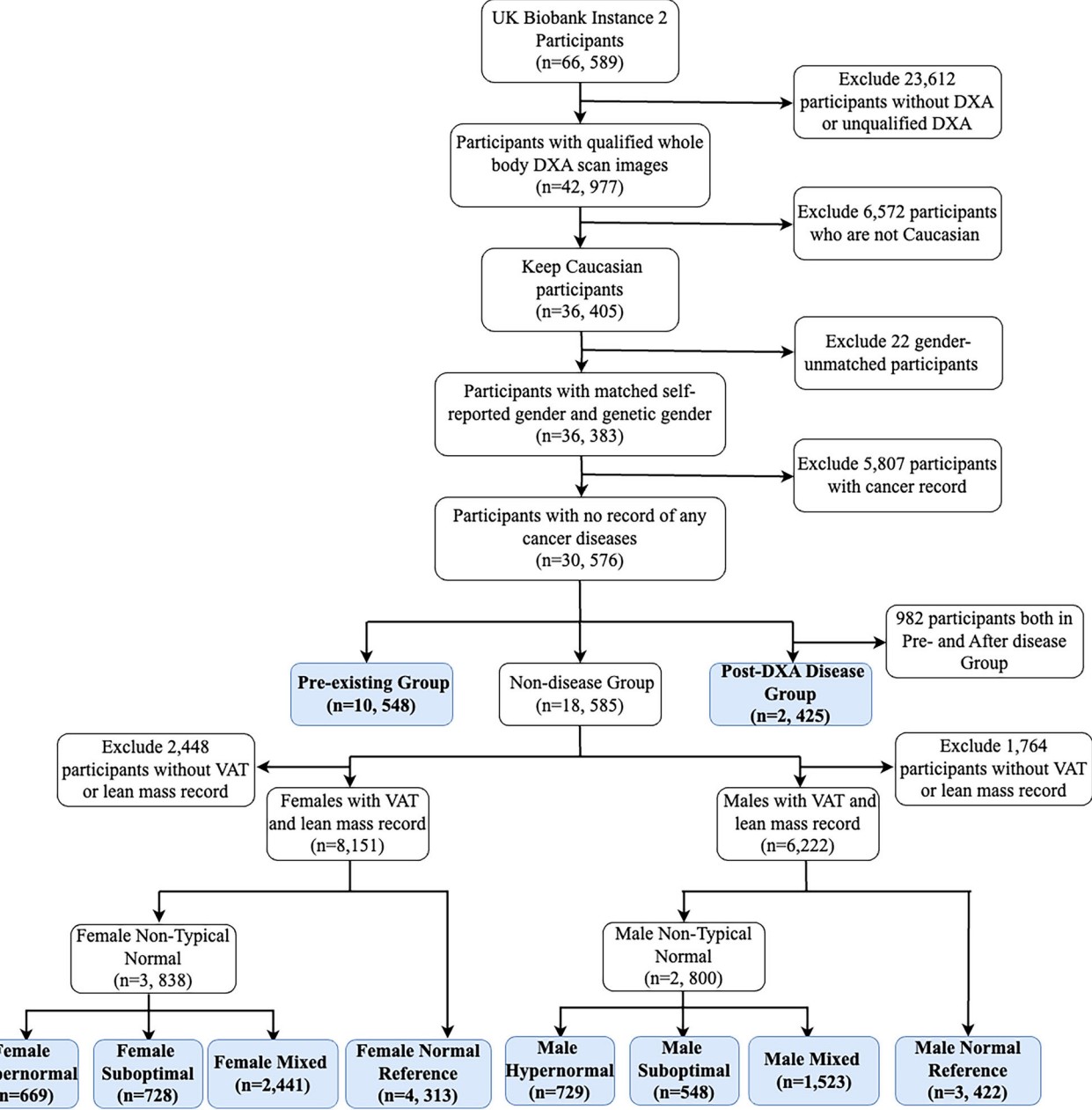

**Fig. 1 | Flowchart of participant selection and inclusion criteria.** Participants flowchart with detailed selection criteria. DXA dual-energy X-ray absorptiometry, VAT visceral adipose tissue.

individual's BCBA. Specifically, during the training process, we use normal reference group participants' DXA image (please refer to Supplementary Method 3 for UK Biobank DXA imaging protocol) as our model's input and use their chronological age as output, i.e., we assume that normal participants' BAs are equal to their CAs, capturing typical ageing patterns in body composition (refer Fig. 2a for details).

To facilitate model training and evaluation, we randomly partitioned the normal reference group dataset into three sets: training (75%), validation (12.5%), and test (12.5%). Each participant contributed only one scan, which simplifies the data division process. To prevent data leakage, we ensured that each scan was assigned exclusively to one of the three sets, maintaining the integrity of the training, validation, and test datasets. Please refer to Supplementary Fig. 1 for the training, validation, and test loss.

However, to identify the most effective architecture, we evaluated ResNet-50, VGG-16, and a Vision Transformer model, all pretrained on ImageNet. Additionally, we compared a voting ensemble model that combines ResNet-50, VGG-16, and DenseNet121. DenseNet121 out-performed all baseline models, and the results can be found in Supplementary Table 1.

### Technical Parameters of Model Training
For each participant, we pre-processed the bone and muscle-fat images by centre cropping and padding into a square shape, ensuring uniformity in input dimensions. Subsequently, both images were resized to 224×224 pixels and concatenated. The concatenated image was then normalised to enhance model performance and convergence.

In terms of hyperparameter optimization, we conducted a comprehensive search that included the following parameters: learning rates of [0.01, 0.001, 0.0001, 0.00001], weight decay rates of [0.000001, 0.00001, 0.0001], and the number of epochs set to [100, 200, 300, 400, 500]. We performed a total of 60 trials using Adam optimisation, allowing us to systematically evaluate the impact of these different hyperparameters.

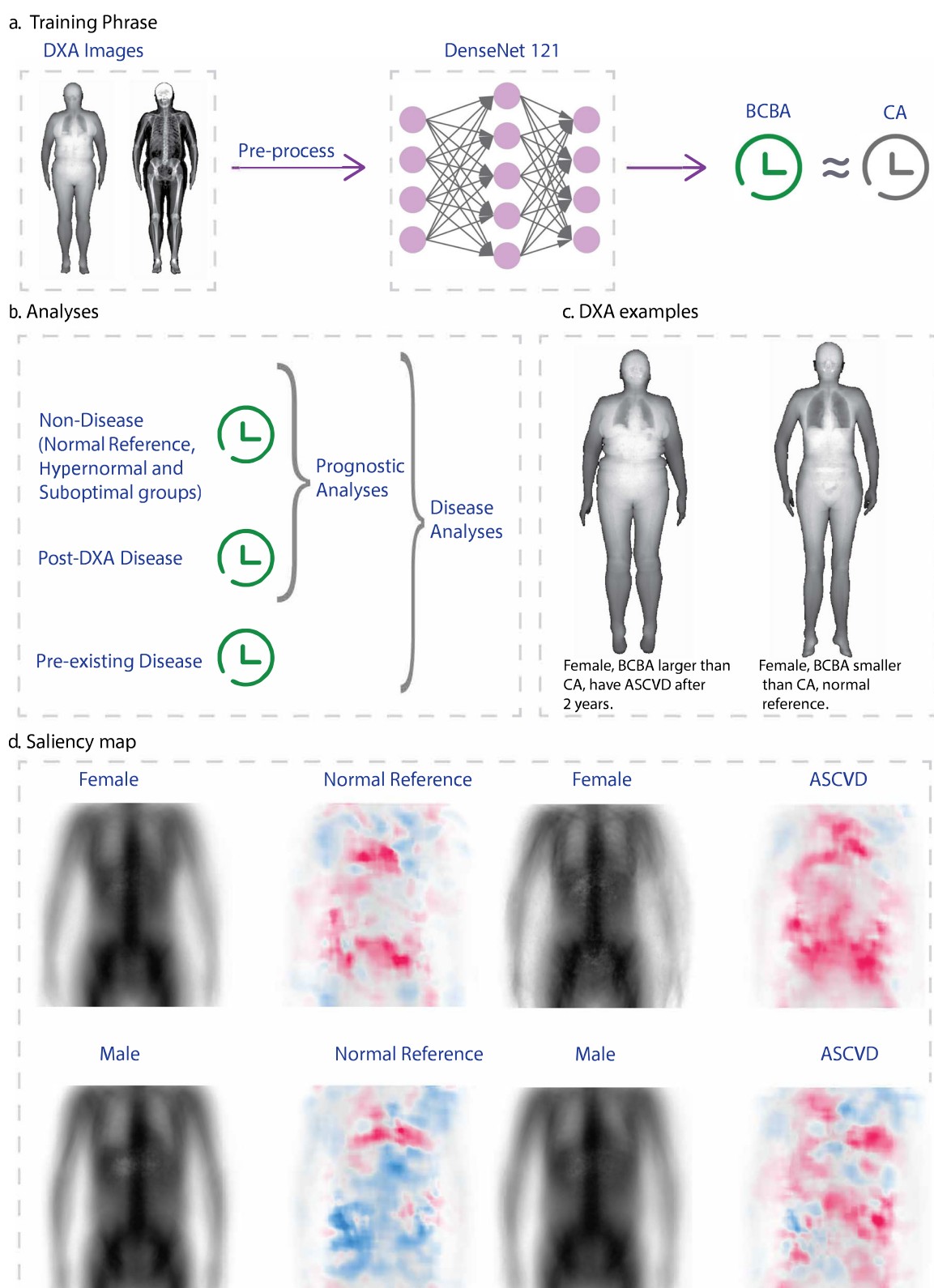

**Fig. 2 | Overview of BCBA model development and validation. a** Overview of training BCBA. The DXA images were processed and put into DenseNet-121. We used the normal reference group's CA as their BCBA. **b** Analyses design for disease biomarker analyses and prognostic analyses. **c** Examples of BCBA larger and smaller than CA participants. **d** Saliency map of BCBA model of the normal reference and pre ASCVD groups. DXA dual-energy X-ray absorptiometry, VAT visceral adipose tissue, BCBA body composition biological age, CA chronological age., ASCVD atherosclerotic cardiovascular disease.

To assess model performance, we utilised metrics including MSE, MAE, R score and MAPE to quantify the disparity between predicted and actual chronological ages, thereby providing insights into the model's accuracy and robustness. Early stopping was included to avoid overfitting. Specifically, training was stopped when the validation loss did not improve for 15 epochs. The final model achieved optimal performance with a learning rate of 0.001, a weight decay of 0.00001, and 300 epochs, using a batch size of 64 on a V100 GPU.

The model selection process involved evaluating performance metrics on the validation set and selecting the configuration that yielded the best results. All model development and training processes were implemented using the PyTorch framework. Detailed descriptions of the model development process and the computational environment can be found in Supplementary Method 4.

### Body Composition Ageing Inference

To validate the efficacy of our body composition ageing model, we conducted inference analyses on various subsets of our dataset, including the normal reference test set, the non-typical body composition group, as well as the pre-existing and post-DXA disease groups (please refer to Fig. 2(b)).

To gain deeper insights into the regions of interest driving the BCBA prediction, we employed the Shapley Additive exPlanations (SHAP) framework. SHAP provides a principled method for interpreting the output of models, enabling the generation of saliency maps for participants' DXA scans, and highlighting the pixel-level[37]. To provide a comprehensive overview, we generated saliency maps for the normal reference group and the pre-existing ASCVD group by randomly sampling 200 participants in each group. Subsequently, we averaged these saliency maps of each group to produce an aggregate saliency image, capturing the common features contributing to BCBA predictions and comparing the pattern of important regions.

### Statistics and Reproducibility

During both the model development and inference stages, we computed several metrics including the R score, RMSE, MSE, MAE, and MAPE. These metrics were instrumental in assessing and comprehending the disparities between the model-estimated BCBA and the participants' CA across different age groups. The inferenced BCBA was then estimated as both disease biomarkers and prognostic biomarkers for the diseases of T2DM, MACE, hypertension and ASCVD (please refer to Fig. 2b).

In the disease biomarkers association study, we used logistic regression analysis, taking the difference of BA and CA (BCBA - CA) and CA as predictors, and each pre-existing disease label as the objective variable. CA was included as a predictor in the model to adjust for differences in disease frequency by age. Participants in the pre-existing disease group of each disease were labelled as positive cases, and the other group of patients were labelled as negative cases. Odd ratios and p-values were reported. Similarly, in the prognostic analyses, participants in pre-existing disease were excluded and patients in the post-DXA disease group were labelled as positive cases for each disease. To directly understand BCBA as prognostic biomarkers, we divided participants into low-risk group (BCBA -CA <= threshold) and high-risk group (BCBA -CA > threshold) for each disease, and KM analyses were applied along with log-rank tests were implemented to measure the differences between low- and high-risk groups.

To compare the predictive performance of our proposed BCBA with general cardiovascular risk profiles, we assessed their ability to predict the risk of MACE and ASCVD. For each group, Cox regression models were constructed, incorporating relevant variables including CA, systolic and diastolic blood pressures, BMI, waist circumference, hip circumference, antihypertensive medication use, smoking status, and glucose levels (model 1). CA, VAT and lean mass ratio were used to build a separate model for comparison (model 2). To evaluate model performance, we employed 10-fold cross-validation and calculated Harrell's C-index and 95% confidence interval (CI).

All analyses were implemented using Python, including package of NumPy, Pandas, Statistics, and Lifelines. All codes can be found online (https://github.com/SereneLian/DXA_BCBA)[38].

## Results

### Data

After the selection process, we reached a total of 4313 females and 3422 males in the normal reference group. The average age for females was 61.893, while for males it was 62.390. The disease groups, which included individuals with both pre-existing disease and post-DXA disease groups, showed considerably higher average ages compared to the normal reference group. The average ages for females 65.667 and 65.772, while for males they were 66.294 and 66.162. As shown in Table 1, the pre-existing disease groups demonstrated significantly higher VAT percentages ($F = 1.328\%$ and $M = 2.330\%$) followed by the post-disease group ($F = 1.327\%$ and $M = 2.271\%$). The normal reference group had the lowest VAT percentage ($F = 0.845\%$ and $M = 1.718\%$). Conversely, the pre-existing disease group had the lowest lean mass percentage ($F = 58.843\%$ and $M = 68.231\%$), followed by the post-DXA disease group in both female and male cohorts ($F = 59.304\%$ and $M = 68.433\%$) compared to the reference group ($F = 61.979\%$ and $M = 71.204\%$).

Additionally, it is noteworthy that male groups exhibited a higher prevalence of pre-existing disease cases compared to their female counterparts ($M = 6254$ cases, 40.7%, and $F = 4294$ cases, 59.3%). Moreover, male groups displayed nearly double the number of positive cases for both T2DM and ASCVD compared to female groups.

### DXA ageing values on normal reference, non-typical and disease cohorts

We trained a deep learning model using the healthy reference' DXA (Fig. 2a) and inferenced on different groups (Fig. 2b). To compare the differences in BCBA across various groups, we employed several performance metrics, including mean square error (MSE), mean absolute error (MAE), mean absolute percentage error (MAPE), and the R score. Please refer to Table 2 MAE values and Supplementary Table 2-3 for MAPE and MSE per age group for females and males.

For the female, the R score for the test group reached 0.872, followed by the post-DXA disease group (0.837), whilst the pre-existing (0.815) disease groups exhibited the lowest scores. For the male group, the R score for the test group achieved a score of 0.858, while the disease groups displayed the lowest R scores (0.807 for post DXA and 0.803 for pre-existing disease). Notably, the lowest score overall was observed in the pre-existing disease group, aligning with our assumption that the BCBA of the normal reference group tends to closely align with their chronological age (CA), while the BCBA of the disease group tends to diverge further from their CA. (Please refer to Supplementary Table 4 for all R scores).

When looking at the differences between BCBA and CA (Supplementary Figs. 2 and 3), the normal group tends to have mean BCBA around 0, whereas the suboptimal and diseased groups have mean differences above zero, whereas the hypernormal group have differences below zero. This suggested that suboptimal groups and disease groups tended to have larger BCBA (i.e. accelerated agers) than CA, while the hypernormal group showed a 'decelerated' ageing trend. (Please refer to Fig. 2c as examples).

Upon further comparison across different age groups, we observed that the 60–69-year-old cohort exhibited BCBA values closer to their CA for all groups. In contrast, the 40-49-year-old group demonstrated more variance with CA among all groups except the female suboptimal group. This likely stems from the relatively few training cases available for these two age groups during the model-building process but also reveals that the 40–49-year-old group has the most varied body composition age which may be different from their chronological age.

The saliency map illustrated in Fig. 2d offers valuable insights into BCBA-related distribution in body composition. Upon comparing the female and male groups, distinct patterns of hot regions were observed in both the normal reference and pre-existing ASCVD groups. In the normal

**Table 1 | Demographic characteristics of participants**

| Groups | Female | | | | | Male | | | | |
|---|---|---|---|---|---|---|---|---|---|---|
| | Normal Reference (n = 4313) | Non-Typical (n = 3838) | Pre-existing disease (n = 4294) | Post-DXA disease (n = 946) | P-Values | Normal Reference (n = 3422) | Non-Typical (n = 2800) | Pre-existing disease (n = 6254) | Post-DXA disease (n = 1479) | P-Values |
| Age Mean, years (mean ± std) | 61.893 ± 7.434 | 62.057 ± 7.335 | 65.667 ± 7.112 | 65.772 ± 7.117 | ****c | 62.390 ± 7.674 | 62.638 ± 7.725 | 66.294 ± 7.2 | 66.162 ± 7.126 | **** |
| VAT ratio[a], % (mean ± std) | 0.845 ± 0.345 | 1.043 ± 0.838 | 1.328 ± 0.747 | 1.327 ± 0.762 | **** | 1.718 ± 0.467 | 1.792 ± 1.196 | 2.330 ± 0.971 | 2.271 ± 0.978 | **** |
| Lean ratio[b], % (mean ± std) | 61.979 ± 3.669 | 62.203 ± 9.579 | 58.843 ± 7.222 | 59.304 ± 6.993 | **** | 71.204 ± 3.124 | 71.799 ± 8.82 | 68.231 ± 6.486 | 68.433 ± 6.578 | **** |
| BMI, kg/m² (mean ± std) | 24.700 ± 2.577 | 25.940 ± 5.409 | 27.760 ± 5.397 | 27.483 ± 5.652 | **** | 25.909 ± 2.319 | 26.364 ± 4.47 | 28.125 ± 4.233 | 28.113 ± 4.36 | **** |
| Waist Circumference, cm, (mean ± std) | 78.867 ± 7.477 | 81.429 ± 13.296 | 86.711 ± 12.869 | 86.158 ± 13.174 | **** | 90.826 ± 6.464 | 91.612 ± 12.405 | 97.308 ± 11.278 | 97.155 ± 11.687 | **** |
| Hip Circumference, cm, (mean ± std) | 98.286 ± 6.198 | 100.471 ± 11.149 | 103.639 ± 10.970 | 103.277 ± 11.540 | **** | 99.02 ± 5.367 | 99.566 ± 7.899 | 102.268 ± 7.869 | 102.592 ± 8.375 | **** |
| T2DM | / | / | 342 | 126 | / | / | / | 742 | 273 | / |
| ASCVD | / | / | 875 | 387 | / | / | / | 1666 | 731 | / |
| MACE | / | / | 555 | 300 | / | / | / | 1505 | 647 | / |
| Hypertension | / | / | 3635 | 464 | / | / | / | 5329 | 591 | / |

[a] VAT ratio was calculated by the mass of VAT divided by total fat and muscle mass; [b] Lean ratio was calculated by the mass of total lean divided by total fat and muscle mass; [c] **** means p <0.0001.

reference group, hot regions (indicating increased BCBA) were predominantly concentrated around the pelvis and lower abdomen in females, whereas in males, the focus was primarily on the chest region alone. Conversely, in the disease group, females exhibited hot regions distributed throughout the upper body, while males displayed a greater emphasis on the chest and lower abdomen regions.

### BCBA can be regarded as multi-disease biomarkers
Based on the assumption that the predicted BCBA of the normal reference group tends to closely align with their CA, while the BCBA of the disease group exhibits greater deviation from their CA, we proceeded to investigate the potential of BCBA as disease biomarkers. As depicted in Fig. 3a, b, our analysis revealed a significant positive association between the age difference of BCBA and CA and the incidence of various diseases.

Specifically, we observed a positive association between the age difference between BCBA and CA and the incidence of ASCVD with an odd ratio of 1.07 ($p < 0.0001$) for females and 1.10 for males ($p < 0.0001$), MACE with an odd ratio of 1.10 for females ($p < 0.0001$) and 1.11 for males ($p < 0.0005$), T2DM with an odd ratio of 1.08 for females ($p < 0.0001$) and 1.04 for males ($p < 0.0001$), and hypertension with an odd ratio of 1.06 for females ($p < 0.0001$) and 1.04 for males ($p < 0.0001$). Please also refer to Table 3 for details. This finding suggests that as the value of BCBA– CA increases, the likelihood of disease incidence also rises, indicating the potential utility of BCBA as a biomarker for various diseases.

### BCBA can be regarded as prognostic biomarkers for multi-disease
To enhance the practical utility of BCBA as a predictive tool, we stratified all participants into low-risk and high-risk groups based on the difference between BCBA and CA (BCBA– CA). This categorisation mirrors conventional ageing definitions, where individuals with BCBA– CA less than or equal to 0 are considered low-risk, while those with BCBA– CA greater than 0 are deemed high-risk. As depicted in Fig. 3c, our analysis revealed a significant differentiation between the high-risk and low-risk groups for ASCVD, MACE, and hypertension in both females and males ($p < 0.005$). Regarding T2DM, we observed a significant distinction between the low-risk and high-risk groups in the female cohort ($p < 0.005$), and a notable separation was observed in the male cohort, reaching statistical significance ($p < 0.05$).

### Comparing BCBA and a general cardiovascular risk profile for ASCVD and MACE prediction
In the analyses of comparing the performance of the BCBA with well-established risk profiles, we conducted a series of experiments to assess their predictive capabilities. Due to the absence of blood test results in the cohort, we included CA, waist circumference, hip circumference, systolic and diastolic blood pressures, smoking status, and BMI for the first model (Model 1). Additionally, CA, VAT ratio and lean ratio rate were used to build a general body composition model (Model 2). Finally, we build our Model 3 by only taking BCBA and the difference between BCBA and CA as inputs. Antihypertensive medication use was excluded in Model 1 due to non-usage among all participants, and glucose levels were omitted due to a high percentage of missing data.

Figure 3d presents the results of our comparison, indicating that Model 3 outperformed Models 1 and 2 for both ASCVD and MACE, indicating that, our model has better performance than directly using body composition data and general risk profiles. For females, our proposed model achieved the highest C-index for both ASCVD (0.679; 95% CI: 0.643–0.710) and MACE (0.678; 95% CI: 0.666–0.691), outperforming demographic-based models (Model 1: ASCVD = 0.656; MACE = 0.645) and body composition model (Model 2 ASCVD = 0.631; MACE = 0.610). When combined with demographic features (Model 1 + 3), our model further improved predictive accuracy, achieving a C-index of 0.684 (ASCVD) and 0.689 (MACE) in females, the best performance across all model combinations.

**Table 2 | Prediction of BCBA vs CA per age group in terms of MAE**

| MAE | Group | Overall | 40-49 | 50-59 | 60-69 | 70-99 |
|---|---|---|---|---|---|---|
| Female | Normal Reference Test Set | 2.883, (2.736, 3.030) | 3.803, (2.968, 4.639) | 2.912, (2.671, 3.153) | 2.609, (2.398, 2.820) | 3.319, (2.934, 3.703) |
| | Hypernormal | 3.147, (2.964, 3.329) | 3.935, (3.368, 4.503) | 3.165, (2.889, 3.441) | 2.898, (2.621, 3.174) | 3.142, (2.418, 3.867) |
| | Suboptimal | 3.281, (3.097, 3.465) | 4.064, (2.880, 5.247) | 3.133, (2.846, 3.419) | 2.964, (2.714, 3.213) | 4.406, (3.836, 4.975) |
| | Pre-existing Disease | 3.291, (3.216, 3.365) | 5.128, (4.487, 5.768) | 3.452, (3.292, 3.613) | 2.963, (2.867, 3.060) | 3.602, (3.449, 3.756) |
| | Post-DXA Disease | 3.075, (2.923, 3.228) | 5.135, (4.328, 5.943) | 3.537, (3.171, 3.902) | 2.722, (2.520, 2.923) | 3.093, (2.804, 3.382) |
| Male | Normal Reference Test Set | 3.206, (3.030, 3.382) | 4.587, (3.583, 5.591) | 3.187, (2.877, 3.496) | 2.908, (2.665, 3.151) | 3.497, (3.090, 3.904) |
| | Hypernormal | 3.349, (3.166, 3.532) | 4.689, (3.737, 5.64) | 3.576, (3.283, 3.869) | 2.823, (2.575, 3.072) | 3.541, (3.034, 4.048) |
| | Suboptimal | 3.354, (3.129, 3.579) | 5.127, (3.840, 6.415) | 3.245, (2.847, 3.642) | 2.888, (2.590, 3.187) | 4.044, (3.517, 4.571) |
| | Pre-existing Disease | 3.409, (3.344, 3.473) | 5.083, (4.436, 5.731) | 3.749, (3.598, 3.901) | 3.062, (2.978, 3.146) | 3.606, (3.485, 3.728) |
| | Post-DXA Disease | 3.320, (3.189, 3.451) | 4.789, (3.734, 5.844) | 3.372, (3.073, 3.671) | 3.058, (2.886, 3.23) | 3.600, (3.338, 3.862) |

Similarly, for males, Model 3 outperformed Model 1 (ASCVD = 0.643 vs. 0.632; MACE = 0.641 vs. 0.639) and Model 2 (ASCVD = 0.604; MACE = 0.635). The integration of our model with Model 1 (Model 1 + 3) demonstrated the most substantial improvement in predictive power, achieving a C-index of 0.703 (ASCVD) and 0.686 (MACE). For the specific values, please refer to Supplementary Table 5.

## Discussion

In this study, we designed BCBA, a DXA scans-based body composition biological age, using a deep learning model, demonstrating its potential as a biomarker of ageing. The model's performance metrics, including the R score, MSE, MAE, and MAPE, on the normal reference group, non-typical group and disease groups, support the assumption that the predicted BCBA of the normal reference group tends to closely align with CA, while the BCBA of the disease group exhibits greater deviation from their CA. This assumption reveals the potential of BCBA as a disease and prognostic biomarkers. As far as we know, this is the first concept for biological ageing using DXA-based body composition.

It is worth highlighting that our analysis design was intentionally separated by sex due to the observed gender-specific differences. There is a significant difference in body composition between male and female groups, such as VAT and lean mass (see Table 1). In addition, we observed a correlation between the frequency of disease occurrences and gender, highlighting the need for future research into sex-specific risk factors and biomarkers.

Our AI approach generates saliency maps that demonstrate hot regions in the diseased cohorts particularly in the abdomen and pelvic regions which makes intuitive sense. There are also differences in males vs females in terms of regions, which aligns with previous research that females tend to accumulate more adipose tissue in the upper body, including the pelvis and abdomen regions, which can contribute to an increased risk of cardiovascular disease[39].

Our disease analyses highlighted BCBA as a potential indicator of our proposed 4 diseases, potentially enabling more targeted preventive strategies and interventions in clinical practices. Additionally, KM analyses underscored the potential of BCBA as a practical tool for prognosticating MACE, ASCVD, T2DM and hypertension. Stratifying individuals based on their BCBA relative to their CA allows clinicians to identify those at higher risk for certain diseases and tailor preventive strategies. Our subanalyses, which compare BCBA with the general cardiovascular risk profile, indicate the potential application of our model which outperformed simply using traditional clinical metrics. Our model (only using BCBA and CA) demonstrated superior performance compared to the general traditional cardiovascular risk model and DXA-derived VAT-lean mass values for MACE and ASCVD. This means that the deep learning approach with the direct use of DXA image derives superior performance compared to just using the DXA-derived VAT and lean mass alone. Importantly, the combined models of our BCBA and clinical risk factors (Model 1 + 3) yielded

the highest predictive performance across both sexes for ASCVD and MACE, underscoring the complementary nature of our approach. These results emphasize that integrating advanced deep learning models with traditional risk factors can enhance predictive accuracy in cardiovascular disease outcomes.

Incorporating this BCBA model into routine clinical practice offers promising potential for enhancing chronic disease risk assessment. The model could integrate into screening processes by providing healthcare providers with a quantitative tool to evaluate patients' chronic health alongside traditional risk factors. This integration could facilitate early identification of at-risk individuals and enable tailored interventions. However, several barriers may hinder its adoption, including the need for healthcare professionals to be trained in interpreting BCBA results, implementation barriers and the necessity for standardised protocols for DXA scans across different clinical settings. Additionally, while the BCBA shows strong correlations with chronic diseases, its actual real-world performance warrants further prospective validation.

There are a few notable limitations in our study. First, while our findings suggest a potential utility of BCBA as a disease and prognostic biomarker, further validation using an external dataset and prospective validation are desirable to confirm these associations and prognostication. Second, it is important to acknowledge that our model was developed and evaluated using data from individuals of Caucasian ethnicity only. Consequently, the generalisability of our findings to different racial and ethnic populations remains to be tested. Future research should aim to assess the applicability and performance of BCBA across diverse racial and ethnic groups, ensuring equitable and inclusive healthcare practices. Moreover, our study only focused on four diseases namely ASCVD, T2DM, MACE, and hypertension. While these diseases are clinically prevalent, the inclusion of additional diseases in future studies would provide a more comprehensive understanding of the utility of BCBA as a biomarker. Future studies could also consider a broader spectrum of diseases, such as cancer and dementia. Besides, in this study, we constructed the normal reference cohorts based solely on VAT and lean mass. However, ageing encompasses a wide range of physiological changes which may influence body composition and health outcomes. To address this limitation, we aim to develop fully automatic methods in the future that do not rely on human-defined selection criteria to determine the normal reference and explore the important features with data-driven methods. Lastly, although we controlled for several known variables, there are likely other confounding factors that were not accounted for in our analysis. Lifestyle variables such as physical activity levels, dietary habits, smoking status, and socioeconomic factors can significantly influence health outcomes, and owing to the paucity of data were not accounted for in our analysis.

In conclusion, we created a sex-specific body composition-based biological ageing, named BCBA, which can be used as an indicator for age-related diseases and prognostic biomarkers.

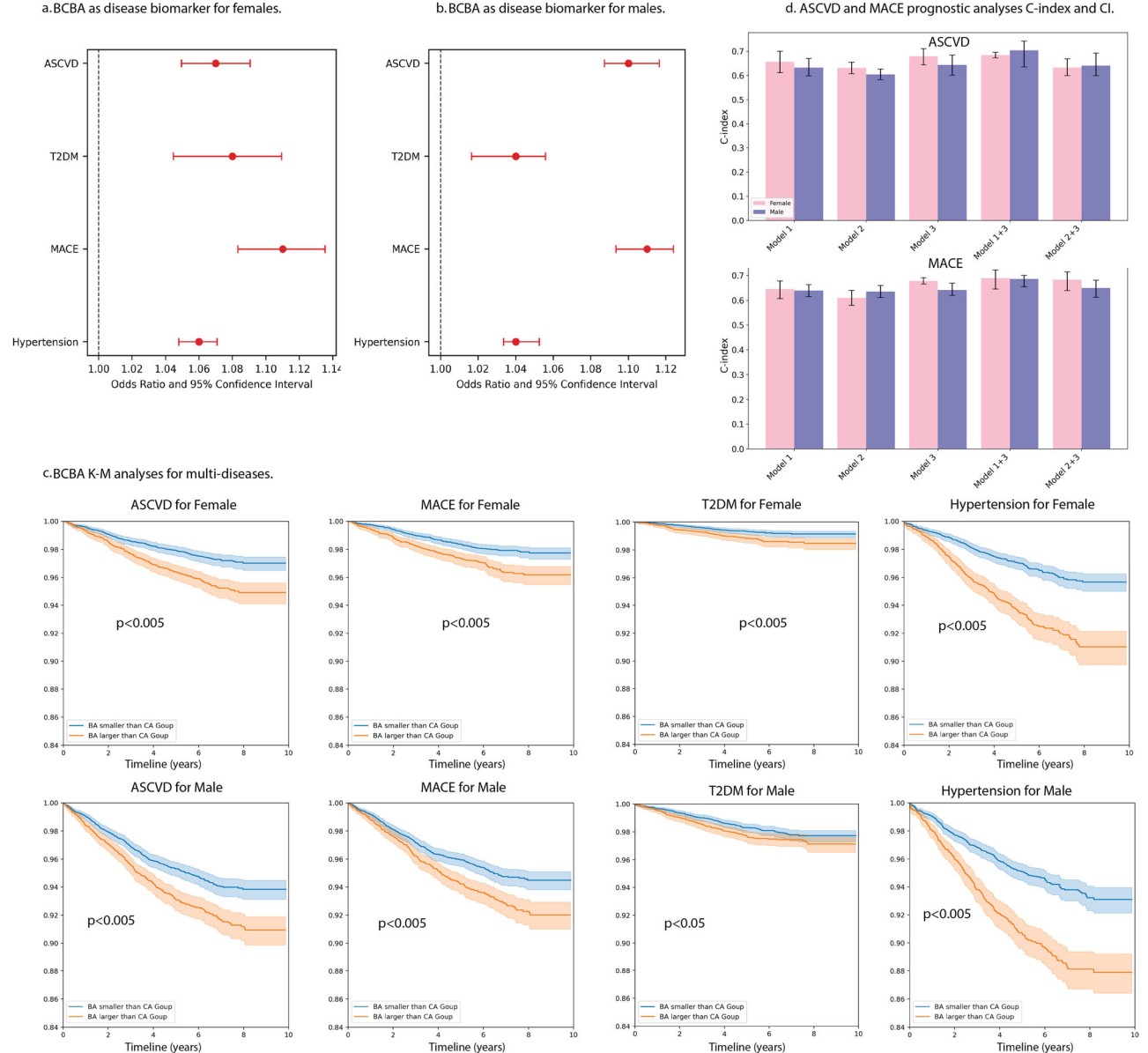

**Fig. 3 | Association and prognostic value of BCBA-CA across disease outcomes.**
**a** Female odd ratios and 95% CI plot of BCBA-CA as disease biomarkers for ASCVD, MACE, T2DM and Hypertension. Red colour means statistically significant.
**b** Female odd ratios and 95% CI plot of BCBA-CA as disease biomarkers for ASCVD, MACE, T2DM and Hypertension. **c** KM curve with 95% CI of BCBA-CA for prognostic analyses with threshold 0. The blue line means the group of participants

with smaller BCBA than CA, while the orange line means the group of participants with larger BCBA than CA. **d** C-index with a 95% CI comparison of Model 1–3 for both females and males. BCBA body composition biological age, ASCVD athero-sclerotic cardiovascular disease, MACE major adverse cardiovascular events, T2DM type 2 diabetes mellitus.

**Table 3 | Odd ratio and 95% CI of BCBA-CA for the incidence of ASCVD, MACE, T2DM and hypertension**

| Sex | Disease | Pvalue | Odd Ratio | 2.50% | 97.50% |
|---|---|---|---|---|---|
| Female | ASCVD | <0.0001 | 1.07 | 1.049 | 1.090 |
| | T2DM | <0.0001 | 1.08 | 1.044 | 1.109 |
| | MACE | <0.0001 | 1.11 | 1.083 | 1.135 |
| | Hypertension | <0.0001 | 1.06 | 1.047 | 1.070 |
| Male | ASCVD | <0.0001 | 1.10 | 1.087 | 1.116 |
| | T2DM | <0.001 | 1.04 | 1.016 | 1.055 |
| | MACE | <0.0001 | 1.11 | 1.093 | 1.123 |
| | Hypertension | <0.0001 | 1.04 | 1.033 | 1.052 |

## Data availability
Authors are not permitted to share the raw data as this is governed by the agreement with the UK Biobank. For this study, permission to access and analyse the UK Biobank data was approved under application number 78730. Other data are available from the corresponding author on reasonable request.

## Code availability
All codes can be found online (https://github.com/SereneLian/DXA_BCBA), or on Zenodo[38].

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

## Acknowledgements

There is no funding for this project. VV is a co-founder of Snowhill Science Ltd. He does not receive a salary from the company. FH is currently an employee of Snowhill Science Ltd, but during this work was hired as a postdoctoral fellow at the University of Hong Kong.

## Author contributions

J.L. contributed on conceptualisation, study design, data analysis, implementation of the computer code, and wrote the manuscript. P.C. contributed on data analysis and revised the manuscript. F.H. contributed on data analysis and wrote the manuscript. J.H. contributed on supervision and revised the manuscript. V.V. contributed on conceptualisation, study design, data analysis, supervision and wrote the manuscript. All authors contributed to the interpretation of the results and approved the final version of the manuscript for submission.

## Competing interests

The authors declare no competing interests.
