## [Transparent Peer Review file · Communications Medicine]

A deep learning sex-specific body composition ageing biomarker using DXA scan

Corresponding Author: Dr Varut Vardhanabhuti

Version 0:

Reviewer comments:

Reviewer #1

(Remarks to the Author)

Summary

The authors present a model which predicts biological age from total-body dual-energy X-ray absorptiometry scans, for use in determining risk of diseases related to aging. The authors examine the association of their predicted biological age to type 2 diabetes mellitus (T2DM), atherosclerotic cardiovascular disease (ASCVD), hypertension, and major adverse cardiovascular events (MACE). The authors use data from the UK Biobank to develop a deep learning model and test it on both normal reference participants and those falling both above (hypernormal) and below (mixed and suboptimal) normal ranges for visceral adipose tissue (VAT) and lean mass derived from DXA. The authors find that their predicted biological age, along with chronological age, is predictive of ASCVD, T2DM, MACE, and hypertension in their cohort, showing modest odds ratios.

General Impressions

The authors present a generally well-formed manuscript on an interesting topic. There are several oversights which prevent recommendation for publication without edits.

Major Comments

The related works section in the introduction to the manuscript is noticeably underdeveloped. Saying that body composition is related to “certain chronic diseases” supported by two references is not sufficient. Especially when one of those references is based on the same data which are being analyzed in this work; is this a general result or not?

Why were those in the pre- and post-disease groups not divided into the normal/hypernormal/mixed/suboptimal groups?

Especially those participants who go on to develop disease post-DXA could be examined along both of these lines. The exclusion of those who do not go on to develop disease limits the predictive power which can be examined from this method.

There is a significant lack of detail in the description of the deep learning training process. The complete hyperparameter search space needs to be described, including the number of trials and method of optimization.

More information is needed on the split of the data into training/testing/validation. How was the data split? Are there more than one scan per participant? If so, how were these split into the different divisions? How did the authors ensure there was no data leakage between the sets? It's unclear which set descriptive statistics are reported on.

The connection to aging is not well-defined. There are many effects in the body due to age such as posture and bone density. However, the association between lean mass and VAT and age has not been found to be significant in the literature. No hypothesis is made about the other associations in the body which could be contributing to this signal, outside of body shape and composition.

Comparisons to models with standard variables, such as demographics, are warranted. There is no reference to any other age acceleration method on this dataset which the authors could compare their method to.

Another valuable comparison would be combining models 1 + 3 and 2 + 3 to see if the imaging-derived marker contributes significant complimentary information to standard markers.

Minor Comments

The odds ratios are a major result and should be reported in the main text.

Supplemental Figure 1 is missing.

Reference 22 does not support the statement it is cited after. Crohn's disease is not a cardiometabolic condition.

The choice to limit the study to only Caucasian participants due to “confounding variables related to genetic diversity” needs to be supported. Have previous studies on body composition measured by DXA shown genetic confounders which would disrupt this analysis?

All Kaplan-Meier curves presented should be on the same scale.

There is no comparison to how AI is helping in this study or how is this different from normally doing the biological age without the AI.

The conflict of interest statement may need to be updated. The Snowhill Sciences Ltd that two authors Fan Huang and Varut Vardhanabhuti are associated with deals with “high-end diagnostics in the field of wellness and longevity, we focus on a data-driven approach utilizing machine learning and AI for accurate prediction.” <https://www.snowhillscience.com/>

Comments to the Editor

We do not recommend this work for publication at this time. If our comments are addressed, we would consider them for further review.

I want to acknowledge that I was aided in this review by the following staff and/or graduate students. I would appreciate it if you would mark them as reviewers and send them a "thank you" email. We are tracking our reviews on the site Publons.com.

Arianna Bunnell (abunnell@hawaii.edu)
Nusrat Zaman Zemi (nusratz@hawaii.edu)
Dustin Valdez (dustinkv@hawaii.edu)

Reviewer #2

(Remarks to the Author)

This manuscript presents a study on the development of a deep learning-based body composition biological age (BCBA) model using DXA scans, trained on data from the UK Biobank. The study explores the relationship between body composition and chronic diseases, proposing BCBA as a potential diagnostic and prognostic biomarker for diseases such as type 2 diabetes mellitus (T2DM), atherosclerotic cardiovascular disease (ASCVD), hypertension, and major adverse cardiovascular events (MACE).

Major Issues:

Training of the DL model is performed using UK Biobank cohort (Caucasian only). The study relies heavily on internal validation using the UK Biobank dataset. However, external validation on an independent dataset is essential to assess the generalizability of the model. At least, validation across diverse populations to ensure the model's applicability in different demographic groups is necessary.

DenseNet-121 was selected as the transfer learning model in this study. Other alternative architectures such as ResNet, Inception, or VGG should be considered and compared to determine the most effective model. Also, it would be interesting to see that how ensemble methods combining multiple pre-trained models or transfer learning techniques improves prediction.

No baseline CNN model been developed and evaluated in comparison to the DenseNet-121 model. Establishing a baseline model would provide a clear benchmark and help determine if the use of a more complex architecture like DenseNet-121 significantly improves performance.

The manuscript does not provide sufficient detail on how class imbalance (e.g., between healthy and diseased individuals) was handled during model training.

The study focuses on four diseases (T2DM, ASCVD, hypertension, and MACE). While these are important, the inclusion of additional age-related diseases such as cancer, dementia, or osteoporosis could provide a more comprehensive evaluation of the BCBA model's utility. The manuscript should discuss the rationale for the selected diseases.

The manuscript does acknowledge some limitations, such as the lack of external validation and the focus on a specific ethnic group. However, it would benefit from a more in-depth discussion of potential confounding factors (e.g., lifestyle variables not accounted for) and the model's limitations in clinical practice/ outside the research setup.

Also, the study could further explore the practical application of BCBA in clinical settings. For instance, how would the model be integrated into routine screening processes? What are the potential barriers to adoption by healthcare providers?

Minor issue:

To ensure that the model is not overfitting, loss plot should be provided.

Reviewer #3

(Remarks to the Author)

Title: Body Composition Biological Age – A deep learning sex-specific body composition aging biomarker using DXA scan

Summary

The authors present a model which predicts biological age from total-body dual-energy X-ray absorptiometry scans, for use in determining the risk of diseases related to aging. The authors examine the association of their predicted biological age to type 2 diabetes mellitus (T2DM), atherosclerotic cardiovascular disease (ASCVD), hypertension, and major adverse cardiovascular events (MACE). The authors use data from the UK Biobank to develop a deep learning model and test it on both normal reference participants and those falling both above (hypernormal) and below (mixed and suboptimal) normal ranges for visceral adipose tissue (VAT) and lean mass derived from DXA. The authors find that their predicted biological age, along with chronological age, is predictive of ASCVD, T2DM, MACE, and hypertension in their cohort, showing modest odds ratios.

General Impressions

The authors present a generally well-formed manuscript on an interesting topic. There are several oversights which prevent recommendation for publication without edits.

Major Comments

The related works section in the introduction to the manuscript is noticeably underdeveloped. Saying that body composition is related to "certain chronic diseases" supported by two references is not sufficient. Especially when one of those references is based on the same data which are being analyzed in this work; is this a general result or not?

Why were those in the pre- and post-disease groups not divided into the normal/hypernormal/mixed/suboptimal groups? Especially those participants who go on to develop disease post-DXA could be examined along both of these lines. The exclusion of those who do not go on to develop disease limits the predictive power which can be examined from this method.

There is a significant lack of detail in the description of the deep learning training process. The complete hyperparameter search space needs to be described, including the number of trials and method of optimization.

More information is needed on the split of the data into training/testing/validation. How was the data split? Are there more than one scan per participant? If so, how were these split into the different divisions? How did the authors ensure there was no data leakage between the sets? It's unclear which set descriptive statistics are reported on.

The connection to aging is not well-defined. There are many effects in the body due to age such as posture and bone density. However, the association between lean mass and VAT and age has not been found to be significant in the literature. No hypothesis is made about the other associations in the body which could be contributing to this signal, outside of body shape and composition.

Comparisons to models with standard variables, such as demographics, are warranted. There is no reference to any other age acceleration method on this dataset which the authors could compare their method to. Is the addition of imaging adding into the

The manuscript limits the analysis to Caucasian participants, citing "confounding variables related to genetic diversity." This justification is insufficient and potentially problematic. The authors should provide a robust explanation supported by literature on why the exclusion of other ethnic groups was necessary. If genetic diversity poses a significant challenge, the authors should discuss how it can be addressed in future studies. It is crucial to ensure that findings are generalizable to diverse populations. Excluding non-Caucasian participants limits the applicability of the results to other ethnic groups, which is particularly important in studies related to health and disease. Recommend that the authors include a discussion on the limitations of excluding diverse populations and propose a plan for future research to validate the findings across different ethnic groups.

Minor Comments

The odds ratios are a major result and should be reported in the main text.

Supplemental Figure 1 is missing.

Reference 22 does not support the statement it is cited after. Crohn's disease is not a cardiometabolic condition.

The choice to limit the study to only Caucasian participants due to "confounding variables related to genetic diversity" needs to be supported. Have previous studies on body composition measured by DXA shown genetic confounders which would disrupt this analysis?

All Kaplan-Meier curves presented should be on the same scale.

There is no comparison to how AI is helping in this study or how is this different from normally doing the biological age without the AI.

The conflict of interest statement may need to be updated. The Snowhill Sciences Ltd that two authors Fan Huang and Varut Vardhanabhuti are associated with deals with "high-end diagnostics in the field of wellness and longevity, we focus on a data-driven approach utilizing machine learning and AI for accurate prediction." Snowhill Sciences.

Reviewer #4

(Remarks to the Author)
co-review

Reviewer #5

(Remarks to the Author)
co-review

Reviewer #6

(Remarks to the Author)
co-review

Version 1:

Reviewer comments:

Reviewer #1

(Remarks to the Author)

Summary:

The authors improve on many of the issues raised in our previous review. The manuscript is in a good-enough state and we recommend publication once the concerns brought up here are addressed.

Comments:

Line 222 & Figure 3C: “[...] without reaching statistical significance ($p < 0.05$).” $p < 0.05$ is statistically significant but this value is reported in both the text and plot. Even if the actual value is > 0.05 , the exact value should be provided, if it is not, the text needs to be changed to reflect that.

In Line 329 the authors state that they train a DenseNet121 for finding BCBA. However, in line 343, they say they use a 2-layer CNN as their baseline. If a 2-layer CNN was used to compare the complex ImageNet architectures against, this is an unfair comparison and needs to be justified.

Lines 337ff.: “Each participant contributed only one scan [...] To prevent data leakage, we ensured that all scans from a single participant were assigned exclusively to one of the three sets”. This is contradictory, either each participant either contributes only one scan or special care has to be taken to not put multiple scans from the same person in different splits.

Line 361f.: “Early stopping was included to avoid overfitting” What was the early stopping criterion?
The cited github repo is not publicly available so it's not possible to look at the code and test for reproducibility.

General: A baseline model that is not sex-specific would add interesting analysis to the text. The authors justify the choice of splitting based on sex but also mention the ability of deep learning to disentangle complex non-linear relationships in the data so having some experiments to justify going with the sex-specific approach would bolster the results presented.

Supplement 2 & 3: Should add the number of people (N) in each group to the table.

Supplement 7: This supplement is sufficiently explored in the main text. The fact that this model adds significant information orthogonal to demographics and the combination of the two is the overall best-performing model present should be mentioned in the paper and sufficiently explored.

Supplement 12 is underdeveloped. “Added an external layer for the regression task” what activation function? Were the targets normalized, if so, how? What loss was used for training, MAE or MSE? For image processing, the images were concatenated along which axis? The channel axis?

Reviewer #2

(Remarks to the Author)

The revised manuscript is recommended for publication.

Version 2:

Reviewer comments:

Reviewer #1

(Remarks to the Author)

Thank you for the revisions. At this point, the manuscript is fit for publication in my opinion. There are some minor issues that could be fixed for a polished final version but they are not critical.

Minor comments:

- The github repo is still not visible for me.
- Supplement 12: What kind of adaptive pooling? Max or Avg?
- Supplement 12: Using raw age values is not ideal if the model was trained with gradient descent since the large absolute values can lead to problematic gradients. Scaling in the 0,1 range would be better and could lead to improved results.

Rebuttal Letter

We would like to extend our gratitude to the reviewers for their valuable comments and efforts. We have addressed each comment point by point and revised our paper accordingly. Additionally, we would like to clarify three errors from the previous version: we incorrectly stated the number of females and the cross-validation number. There were some typos in Figure 1. All these errors have now been corrected

Reviewer #1 (Remarks to the Author):

Summary

The authors present a model which predicts biological age from total-body dual-energy X-ray absorptiometry scans, for use in determining risk of diseases related to aging. The authors examine the association of their predicted biological age to type 2 diabetes mellitus (T2DM), atherosclerotic cardiovascular disease (ASCVD), hypertension, and major adverse cardiovascular events (MACE). The authors use data from the UK Biobank to develop a deep learning model and test it on both normal reference participants and those falling both above (hypernormal) and below (mixed and suboptimal) normal ranges for visceral adipose tissue (VAT) and lean mass derived from DXA. The authors find that their predicted biological age, along with chronological age, is predictive of ASCVD, T2DM, MACE, and hypertension in their cohort, showing modest odds ratios.

General Impressions

The authors present a generally well-formed manuscript on an interesting topic. There are several oversights which prevent recommendation for publication without edits.

Major Comments

1. The related works section in the introduction to the manuscript is noticeably underdeveloped. Saying that body composition is related to “certain chronic diseases” supported by two references is not sufficient. Especially when one of those references is based on the same data which are being analyzed in this work; is this a general result or not?

Thank you for your valuable comments. We have expanded the related works section in the introduction, specifically from lines 49 to 58, by adding additional references to support the connection between body composition and chronic diseases. These references draw on data from various datasets, reinforcing the generality of this conclusion across different cohorts.

2. Why were those in the pre- and post-disease groups not divided into the normal/hypernormal/mixed/suboptimal groups? Especially those participants who go on to develop disease post-DXA could be examined along both of these lines. The exclusion of those who do not go on to develop disease limits the predictive power which can be examined from this method.

Thank you for your valuable comments; this is an important question. We chose not to divide the pre- and post-disease groups into subcategories because our primary goal was to develop a

biological age indicator based only on a healthy population. Our assumption is that individuals in the normal reference group have BA that are aligned with CAs. We then compared the BAs of the pre-disease and post-DXA groups with those of healthy individuals to evaluate the potential of BA as a diagnostic and prognostic biomarker.

Regarding the post-DXA group, we recognise that these participants were healthy at the time of screening, but there may be subtle indicators on the DXA scan that suggest underlying features. Our proposed BA can be explored as a tool to identify these hidden changes, as indicated by our prognostic analyses. Additionally, while we utilised VAT and lean mass as criteria for defining the normal reference group, we did not assume a direct relationship between these parameters and the disease groups. Instead, our assumptions pertain specifically to the BCBA that we have proposed.

Regarding the comment about the exclusion of individuals who do not go on to develop disease, limiting the predictive power of our method, we apologise for any misunderstanding. We interpret your concern as relating to our prognostic analyses. We would like to clarify that our analyses included both healthy individuals and post-DXA patients. We value your comments and will make sure to express this clearly in the manuscript.

3. There is a significant lack of detail in the description of the deep learning training process. The complete hyperparameter search space needs to be described, including the number of trials and method of optimization.

Thanks for your comments, we have elaborated the methods section with training processing including the hyperparameter search space, the number of trials and method of optimization from line 355 to line 364.

4. More information is needed on the split of the data into training/testing/validation. How was the data split? Are there more than one scan per participant? If so, how were these split into the different divisions? How did the authors ensure there was no data leakage between the sets? It's unclear which set descriptive statistics are reported on.

Thank you for your valuable feedback. We appreciate your request for more detailed information regarding the data split for training, testing, and validation. We added detailed data split from line 336 to 341.

To facilitate model training and evaluation, we randomly partitioned the normal reference group dataset into three sets: training (75%), validation (12.5%), and test (12.5%). Each participant has only one scan, which simplifies the data division process. To prevent data leakage, we ensured that all scans from a single participant were assigned exclusively to one of the three sets, maintaining the integrity of the training, validation, and test datasets.

Regarding descriptive statistics, Table 1 presents the combined data for the training, validation, and test sets, providing an overview demographic information of the dataset. Additionally, Table

2 displays the model's prediction performance metrics specifically for the testing set, demonstrating the model's predictive accuracy.

We hope this clarification enhances the manuscript, and we thank you for your constructive comments.

5. The connection to aging is not well-defined. There are many effects in the body due to age such as posture and bone density. However, the association between lean mass and VAT and age has not been found to be significant in the literature. No hypothesis is made about the other associations in the body which could be contributing to this signal, outside of body shape and composition.

Thank you for your valuable feedback on our study's connection to aging. We acknowledge that ageing involves a wide range of physiological changes, including alterations in posture and bone density, which may influence body composition and health outcomes.

In our research, we specifically focused on the relationship between body composition—particularly lean mass and VAT—and biological age as indicated by our BCBA model. We have identified some associations between lean mass, VAT, and age. For instance, a study involving a large cohort from the UK Biobank found that greater VAT mass was significantly associated with older age [4]. Additionally, a study examining body composition across various age groups identified breakpoints in the relationship between body composition parameters (including lean mass and fat mass) and age, suggesting that significant associations do exist and vary across different age cohorts [5]. Furthermore, a comprehensive review of body composition changes with ageing emphasises that both VAT and lean mass are critical components influencing health risks and functional abilities in older adults [6]. Thus, we hypothesise that lean mass and VAT serve as essential markers of health that may more accurately reflect biological age than chronological age alone. To provide a more comprehensive review, we have added the references to the background section from line 77 to line 88.

In our study, we acknowledged the lack of assumptions about other associations with aging as a limitation. In the discussion section from line 227 to line 285, we outline potential mechanisms might relate to ageing and disease risk. In the next work, we aim to develop fully automatic methods in the future that do not rely on human-defined selection criteria to determine the normal reference and use data-driven method to explore those factors automatically to solve this limitation.

In conclusion, although the literature on lean mass, VAT, and age is not extensive, there is supporting evidence for their role in the aging process, and our study aims to build on this by demonstrating the value of BCBA as a diagnostic and prognostic biomarker.

6. Comparisons to models with standard variables, such as demographics, are warranted. There is no reference to any other age acceleration method on this dataset which the authors could compare their method to.

Thank you for your comments. We acknowledge the lack of references to other age acceleration methods in this dataset, primarily due to data limitations. For instance, we used instance 2 dataset of UK Biobank, and this instance does not have blood test samples. To overcome this, we compared our model to general cardiovascular risk profiles, which included chronological age, sex, waist circumference, hip circumference, systolic and diastolic blood pressures, smoking status, and BMI for ASCVD and MACE (lines 224 to 233). Our results indicated that our proposed method outperformed these standard variables, highlighting its potential as a valuable tool in risk assessment.

7. Another valuable comparison would be combining models 1 + 3 and 2 + 3 to see if the imaging-derived marker contributes significant complimentary information to standard markers.

Thank you for highlighting this valuable comparison. We have included the results for models 1 + 3 and 2 + 3 in Supplementary 7. Our findings indicate that combining the models improves prediction performance overall, with model 1 + 3 demonstrating a significantly greater improvement compared to model 2 + 3. This may be attributed to the overlap in information between models 2 and 3, while models 1 and 3 provide complementary information that enhances predictive capability.

Supplementary 7: C-index of model 1-3 comparison for ASCVD and MACE

		ASCVD	MACE
Female	Model 1	0.656 (0.611, 0.700)	0.645 (0.607, 0.678)
	Model 2	0.631 (0.607, 6.654)	0.610 (0.579, 0.640)
	Model 3 [our model]	0.679 (0.643, 0.710)	0.678 (0.666, 0.691)
	Model 1+3	0.684 (0.672, 0.695)	0.689 (0.645, 0.722)
	Model 2+3	0.632 (0.599, 0.668)	0.683 (0.639, 0.714)
Male	Model 1	0.632 (0.598, 0.669)	0.639 (0.615, 0.663)

	Model 2	0.604 (0.582, 0.626)	0.635 (0.611, 0.659)
	Model 3 [our model]	0.643 (0.601, 0.684)	0.641 (0.620, 0.669)
	Model 1+3	0.703 (0.635, 0.741)	0.686 (0.654, 0.700)
	Model 2+3	0.640 (0.599, 0.692)	0.649 (0.612, 0.681)

Minor Comments

1. The odds ratios are a major result and should be reported in the main text.

Thank you for your suggestion. We have included the odds ratios in the main text, presented as Table 3.

2. Supplemental Figure 1 is missing.

We apologize for the oversight regarding the order of the supplementary files. We have re-organized the supplementary materials, and you can now find the figures as Supplementary Figures 5 and 6. Thank you for your understanding.

3. Reference 22 does not support the statement it is cited after. Crohn's disease is not a cardiometabolic condition.

Thank you for your comments. We have modified the statement to clarify the context and have added a new statement to accurately reflect the findings of reference 22. We appreciate your attention to this detail.

4. The choice to limit the study to only Caucasian participants due to “confounding variables related to genetic diversity” needs to be supported. Have previous studies on body composition measured by DXA shown genetic confounders which would disrupt this analysis?

Thanks for raising this important point. We included only Caucasian participants with following concerns supported by previous research:

Thank you for highlighting this important issue. We chose to include only Caucasian participants due to the following concerns, supported by previous research:

- **Genetic Variability and Body Composition:** Research has demonstrated that genetic factors can significantly influence body composition and metabolic health. For example, a study indicated that genetic diversity can affect body fat distribution and lean mass measurements across different populations, suggesting that failing to account for such variability may lead to misleading conclusions in studies utilizing DXA measurements [1].
- **Geographic and Genetic Influences:** Another study emphasized that determinants of body composition can vary geographically due to genetic and environmental factors. This underscores the importance of considering genetic diversity when analysing body composition data, as unaccounted genetic differences could confound results [2].
- **Polymorphisms Affecting Body Composition:** A study investigating genetic polymorphisms related to muscle biology found that variations could influence muscle mass and fat distribution, particularly in older individuals. This highlights the potential for genetic factors to confound analyses of body composition, reinforcing our decision to limit participant diversity to enhance precision in our findings [3].

By restricting our study to a more homogeneous population, we aim to minimize confounding effects related to genetic diversity, thereby enhancing the validity of our results.

5. All Kaplan-Meier curves presented should be on the same scale.

Thank you for your suggestion. We have made all KM curves on the same scale in Figure 3.

6. There is no comparison to how AI is helping in this study or how is this different from normally doing the biological age without the AI.

The AI-based model in our study plays a crucial role in extracting complex patterns from multi-dimensional body composition data. Traditional methods for calculating biological age often rely on simpler models or limited 1D clinical markers (e.g., chronological age, basic health metrics) with linear models, which may overlook subtle interactions between different body composition factors with chronic diseases. By using AI, specifically deep learning techniques, we can process this large 2D **image** dataset (more complex data) and uncover **deep hidden non-linear relationships** that might not be evident through traditional regression or statistical methods. This enables the model to provide a more refined and personalised BA estimate that takes into account more complex biological processes.

Comparison to Traditional Biological Age Calculation:

In traditional approaches to estimating biological age, researchers typically rely on clinical biomarkers or physiological measurements like telomere length, epigenetic clocks, or basic health indicators. These methods, while valuable, often lack the ability to fully capture the dynamic interactions within body composition and fail to integrate multi-dimensional data effectively. Additionally, some methods require expensive data, such as genetic sequencing. In contrast, our AI-driven model offers several advantages:

- Integration of diverse factors: Our AI-driven model leverages the ability to process high-dimensional data, allowing for the integration of diverse hidden variables from DXA scans. This capability enables the model to learn complex patterns that are often missed by traditional methods.
- Improved prediction accuracy: The results indicate that the biological age indicator we developed, BCBA, outperforms standard cardiovascular risk profiles in predicting the onset of MACE and ASCVD. We believe that this has suggested that AI methods can slightly outperform traditional methods.
- Data availability: DXA scans yield accurate results while being time-efficient and cost-effective, making our model suitable for widespread use in potential clinical practices.

7. The conflict of interest statement may need to be updated. The Snowhill Sciences Ltd that two authors Fan Huang and Varut Vardhanabhuti are associated with deals with “high-end diagnostics in the field of wellness and longevity, we focus on a data-driven approach utilizing machine learning and AI for accurate prediction.” <https://www.snowhillscience.com/>

The study was performed with the University of Hong Kong.

Author (VV) is a co-founder of Snowhill Science Ltd, and receives no salary from the company.

Author (FH) worked on this project as a post-doctoral fellow at the University of Hong Kong at the time of the project. He now works at Snowhill Science Ltd.

For transparency, we have updated the affiliation and disclosure statement to reflect the above.

[1] Ciardullo, Stefano, et al. "Differential association of sex hormones with metabolic parameters and body composition in men and women from the United States." *Journal of Clinical Medicine* 12.14 (2023): 4783.

[2] Rathnayake, Nirmala, Hasanga Rathnayake, and Sarath Lekamwasam. "Age-Related Trends in Body Composition among Women Aged 20–80 Years: A Cross-Sectional Study." *Journal of Obesity* 2022.1 (2022): 4767793.

[3] Bashir, Tufail, et al. "Activin type I receptor polymorphisms and body composition in older individuals with sarcopenia—Analyses from the LACE randomised controlled trial." *Plos one* 18.11 (2023): e0294330.

[4] Yu, Bowei, et al. "Age-specific and sex-specific associations of visceral adipose tissue mass and fat-to-muscle mass ratio with risk of mortality." *Journal of cachexia, sarcopenia and muscle* 14.1 (2023): 406-417.

[5] Briand, Marguerite, et al. "Body composition and aging: cross-sectional results from the INSPIRE study in people 20 to 93 years old." *GeroScience* (2024): 1-13.

[6] Looker, Anne C., et al. "Age, gender, and race/ethnic differences in total body and subregional bone density." *Osteoporosis international* 20 (2009): 1141-1149.

Reviewer #2 (Remarks to the Author):

This manuscript presents a study on the development of a deep learning-based body composition biological age (BCBA) model using DXA scans, trained on data from the UK Biobank. The study explores the relationship between body composition and chronic diseases, proposing BCBA as a potential diagnostic and prognostic biomarker for diseases such as type 2 diabetes mellitus (T2DM), atherosclerotic cardiovascular disease (ASCVD), hypertension, and major adverse cardiovascular events (MACE).

Major Issues:

1. Training of the DL model is performed using UK Biobank cohort (Caucasian only). The study relies heavily on internal validation using the UK Biobank dataset. However, external validation on an independent dataset is essential to assess the generalizability of the model. At least, validation across diverse populations to ensure the model's applicability in different demographic groups is necessary.

Thank you for pointing out this important question. The UK Biobank includes several other ethnic groups, but we only included Caucasian individuals because of their consideration of genetic variability and body composition. Research has demonstrated that genetic factors can significantly influence body composition and metabolic health. For example, a study indicated that genetic diversity could affect body fat distribution and lean mass measurements across different populations, suggesting that failing to account for such variability may lead to misleading conclusions in studies utilising DXA measurements.

As the study is a proof of concept in looking at DXA images for BCBA calculation, we plan to validate with other external and ethnically diverse cohorts in the future. Currently, there are no available datasets of reasonable size to test this hypothesis. We are planning future prospective studies to further test validate this proof of concept. We have acknowledged this in the limitation section.

2. DenseNet-121 was selected as the transfer learning model in this study. Other alternative architectures such as ResNet, Inception, or VGG should be considered and compared to determine the most effective model. Also, it would be interesting to see that how ensemble methods combining multiple pre-trained models or transfer learning techniques improves prediction.

Thank you for your thoughtful comments regarding the selection of DenseNet-121 as our transfer learning model. We agree that exploring alternative architectures such as ResNet, Inception, and VGG is valuable for assessing model effectiveness.

In our initial experiments, we also evaluated ResNet-50, VGG-16, and a Vision Transformer model. While DenseNet-121 ultimately achieved the best performance on the test set, ResNet-50 and the Vision Transformer converged quickly and demonstrated strong performance during training. However, they exhibited limitations in generalizability on the test set.

To provide transparency, we have included the performance metrics for ResNet-50, VGG-16, and the Transformer model in Supplementary 11. This should offer additional insights into the

comparative effectiveness of these architectures. All of the aforementioned models were pretrained on ImageNet and we downloaded the pre-trained weights from Pytorch.

As for ensemble models, this is an interesting point. We added a simple voting ensemble method, taking the mean value of pretrained ResNet-50, VGG-16 and DenseNet121 as the final prediction results. The MSE is 14.453 and R score is 0.865.

Supplementary 11: ResNet-50, VGG-16, Vision Transformer and Voting Ensembled model comparing with DenseNet121 on testing set (combine male and female).

Model	MSE	R score
CNN	24.087	0.744
VGG-16	18.411	0.809
ResNet-50	18.016	0.815
Vision Transformer	17.552	0.824
Voting Ensemble model	14.453	0.865
DenseNet121	14.012	0.866

3. No baseline CNN model been developed and evaluated in comparison to the DenseNet-121 model. Establishing a baseline model would provide a clear benchmark and help determine if the use of a more complex architecture like DenseNet-121 significantly improves performance.

Thank you for your valuable suggestion regarding the establishment of a baseline CNN model. We have indeed developed a 2-layer CNN model connected with a fully connected layer as baseline for comparison. We have now included its performance metrics in Supplementary 11, with MSE of 24.087 and R score of 0.744, which provides a benchmark to evaluate the effectiveness of the more complex DenseNet-121 architecture.

4. The manuscript does not provide sufficient detail on how class imbalance (e.g., between healthy and diseased individuals) was handled during model training.

Thank you for your suggestion. We apologise for any misunderstanding. However, we would like to clarify that our model was trained for a regression task. The disease diagnosis aspect, which involves comparing healthy and diseased individuals, is conducted as a clinical association study using statistical methods. Therefore, we do not feel the class imbalance should be an issue as our approach does not involve training a classification model.

5. The study focuses on four diseases (T2DM, ASCVD, hypertension, and MACE). While these are important, the inclusion of additional age-related diseases such as cancer, dementia, or osteoporosis could provide a more comprehensive evaluation of the BCBA model's utility. The manuscript should discuss the rationale for the selected diseases.

Thank you for your insightful feedback regarding the selection of diseases in our study. We focused on T2DM, ASCVD, hypertension, and MACE due to their significant prevalence, large number of positive cases in this dataset, and impact on public health, particularly in relation to

body composition and metabolic health. These conditions are closely linked to cardiovascular health and are often intertwined with one another, making them ideal for evaluating our biological age indicator (BCBA).

However, we acknowledge that including additional age-related diseases such as cancer, dementia, or osteoporosis could enhance the comprehensiveness of our analysis. We did not include other chronic diseases such as dementia and osteoporosis, mainly due to the limitation of positive cases. As for cancer, we are aiming to build a top 20 common cancer-related biomarkers in the next study, since this is an initial study. Moreover, cancers are a group of heterogeneous diseases, with various different causative factors so we chose not to include them owing to potential complexities in subanalysis.

In our revised manuscript, we have added a sentence on the rationale for selecting the initial four diseases (line 136 to 138) and the potential value of expanding the scope to include other age-related conditions in future research (line 298-300).

6. The manuscript does acknowledge some limitations, such as the lack of external validation and the focus on a specific ethnic group. However, it would benefit from a more in-depth discussion of potential confounding factors (e.g., lifestyle variables not accounted for) and the model's limitations in clinical practice/ outside the research setup.

Thank you for your valuable feedback regarding the discussion of limitations in our manuscript. We have revised the limitations section to include additional context about confounding factors and study settings (line 299-303). In response to your subsequent comments, we have also added a new paragraph discussing the potential and limitations of our model in clinical practice, located from line 277 to line 285.

7. Also, the study could further explore the practical application of BCBA in clinical settings. For instance, how would the model be integrated into routine screening processes? What are the potential barriers to adoption by healthcare providers?

Thank you for pointing out this important idea. We have added a new paragraph discussing the potential and limitations of our model in clinical practice, located from line 277 to line 285, with the following content:

“Incorporating this BCBA model into routine clinical practice offers promising potential for enhancing chronic disease risk assessment. The model could be integrated into screening processes by providing healthcare providers with a quantitative tool to evaluate patients’ chronic health alongside traditional risk factors. This integration could facilitate early identification of at-risk individuals and enable tailored interventions. However, several barriers may hinder its adoption, including the need for healthcare professionals to be trained in interpreting BCBA results, implementation barriers and the necessity for standardised protocols for DXA scans across different clinical settings. Additionally, while the BCBA shows strong correlations with chronic diseases, its actual real-world performance warrants further prospective validation.”

Minor issue:

To ensure that the model is not overfitting, loss plot should be provided.

Thank you for your suggestion. The loss plot for train, validation and test data have been added in Supplementary 10.

Supplementary 10: Best performed model training, validation, and test loss

Dear Reviewer,

Thank you for your detailed and thoughtful review of our paper. We have carefully considered your suggestions and made the necessary changes to improve the manuscript. Below, we provide a point-by-point response to address each of your comments:

1. Line 222 & Figure 3C: “[...] without reaching statistical significance ($p < 0.05$).” $p < 0.05$ is statistically significant but this value is reported in both the text and plot. Even if the actual value is > 0.05 , the exact value should be provided, if it is not, the text needs to be changed to reflect that.

Thank you for your detailed review and for pointing out this issue. We apologise for the oversight. The corrected sentence should read: “A notable separation was observed in the male cohort, reaching statistical significance.” The actual p-value for this observation is 0.03. We have deleted the “without” on line 221.

2. In Line 329 the authors state that they train a DenseNet121 for finding BCBA. However, in line 343, they say they use a 2-layer CNN as their baseline. If a 2-layer CNN was used to compare the complex ImageNet architectures against, this is an unfair comparison and needs to be justified.

Thank you for your valuable suggestion. In the previous revision, one of the reviewers recommended including a simple baseline model and several complex model for comparison. To address this, we had quickly trained a 2-layer CNN model. However, we agree that the 2-layer CNN may not provide a fair comparison. Therefore, we have removed the CNN model from both the main manuscript and the supplementary materials.

3. Lines 337ff.: “Each participant contributed only one scan [...] To prevent data leakage, we ensured that all scans from a single participant were assigned exclusively to one of the three sets”. This is contradictory, either each participant either contributes only one scan or special care has to be taken to not put multiple scans from the same person in different splits.

Thank you for your detailed comments. We apologize if the original sentence was unclear. To clarify, each participant has only one scan, and each scan was assigned exclusively to one of the three sets. We have revised the sentence at line 353 to ensure clarity and to avoid potential misunderstandings in the future.

4. Line 361f.: “Early stopping was included to avoid overfitting” What was the early stopping criterion?

Thank you for pointing this out. The early stopping criterion was based on the validation loss. Specifically, training was stopped when the validation loss did not improve for 15 epochs. We have added the stopping criteria at line 376.

5. The cited github repo is not publicly available so it's not possible to look at the code and test for reproducibility.

Thanks for your suggestion. The code now should be public at https://github.com/SereneLian/DXA_BCBA/tree/main

6. General: A baseline model that is not sex-specific would add interesting analysis to the text. The authors justify the choice of splitting based on sex but also mention the ability of deep learning to disentangle complex non-linear relationships in the data so having some experiments to justify going with the sex-specific approach would bolster the results presented.

Thank you for your thoughtful suggestion. We appreciate the idea of including a non-sex-specific baseline model. However, since the inception of this study, we have recognized that sex is a critical factor in body composition analysis. While a non-sex-specific model could provide meaningful insights, it is not compatible with our current design because the normal reference group is defined based on sex-specific factors.

As mentioned in the discussion section, our goal for future work is to develop fully automated methods that do not rely on predefined selection criteria (such as sex-specific models) to establish the normal reference group. Instead, we aim to use data-driven approaches to identify important features without human-imposed constraints.

7. Supplement 2 & 3: Should add the number of people (N) in each group to the table.

Thank you for your detailed suggestion. We have now added the number of people in each group to the table for clarity.

8. Supplement 7: This supplement is sufficiently explored in the main text. The fact that this model adds significant information orthogonal to demographics and the combination of the two is the overall best-performing model present should be mentioned in the paper and sufficiently explored.

Thank you for your valuable feedback. We appreciate your suggestion to emphasize the findings in Supplementary 7 within the main text. To address this, we have expanded the discussion in the main manuscript to highlight that our model adds significant orthogonal information beyond demographics and body composition factors, as demonstrated by the improved performance of the combined models (Model 1+3 and Model 2+3), at line 237 -line 247 and line 285 – line 289.

9. Supplement 12 is underdeveloped. "Added an external layer for the regression task" what activation function? Were the targets normalized, if so, how? What loss was used for training, MAE or MSE? For image processing, the images were concatenated along which axis? The channel axis?

Thanks for your suggestion. We have now changed Supplement 12 as following. As for

targets normalisation, we tried both normalized and non- normalized labels and noticed that there was no difference for model performance.

Supplementary 12: BCBA model development, process and the computational environment

- Model development: We used PyTorch's DenseNet-121 as the backbone, by using the `torch.model.densenet.features` as internal output, then added a fully connected layer with a relu activation function and a adaptive pool layer for the regression task.
- Target: The target values were the raw age value.
- Loss function: Mean Squared Error (MSE) was used as the loss function for training.
- Image processing: The two input images were centre-cropped and normalized. The processed images were concatenated along the channel axis to form a single input image.
- Environment: Python 3.8, Pytorch 1.12, sklearn 1.30, numpy 1.24 and pandas 2.0.3.

Dear Reviewer,

Thank you for your detailed and thoughtful review of our manuscript. We greatly appreciate your time and feedback, which have been invaluable in improving our work. Below are our point-by-point responses and the corresponding updates made to the supplementary materials:

-1. The github repo is still not visible for me.

Thank you for bringing this to our attention. We have resolved the issue with the GitHub repository link, and it is now publicly accessible.

-2. Supplement 12: What kind of adaptive pooling? Max or Avg?

We used average pooling in our method, and we have updated Supplement 12 to reflect this information.

-3. Supplement 12: Using raw age values is not ideal if the model was trained with gradient descent since the large absolute values can lead to problematic gradients. Scaling in the 0,1 range would be better and could lead to improved results.

Thank you for your valuable feedback regarding the use of raw age values as the target for regression. This is a good point. While it is true that large absolute values may lead to problematic gradients in some scenarios when using gradient descent, our experimental setup mitigated this potential issue for the following reasons:

- (1) Adam optimisers, which we used in our experiments, are less sensitive to the scale of the target variable due to their adaptive learning rate mechanisms. This reduces the risk of unstable gradients caused by large absolute values.
- (2) We conducted experiments using both raw age values and normalised (scaled to the [0, 1] range) targets at the very beginning of this study. The results did show significant differences in the model's performance metrics (MSE 13.168 (raw) vs MSE 13.189 for female test group and 15.793 vs MSE 16.011 for male test group, both the best performance), suggesting that the scale of the target did not adversely impact model optimisation in our case. Since we thought this part may be very trivial, we did not add it in the manuscript.
- (3) Using raw age values directly as the target allows for more interpretable model outputs, as the predictions are directly in the unit of interest (years). Normalising the target would require an additional post-processing step to rescale predictions, which could introduce unnecessary complexity.

In this case, we decided to use the raw value instead of normalised value in the final version.